# Influence of Gating System Parameters of Die-Cast Molds on Properties of Al-Si Castings

**DOI:** 10.3390/ma14133755

**Published:** 2021-07-05

**Authors:** Štefan Gašpár, Tomáš Coranič, Ján Majerník, Jozef Husár, Lucia Knapčíková, Dominik Gojdan, Ján Paško

**Affiliations:** 1Department of Technical Systems Design and Monitoring, Faculty of Manufacturing Technologies with a Seat in Prešov, Technical University of Košice, Štúrova 31, 080 01 Prešov, Slovakia; stefan.gaspar@tuke.sk (Š.G.); tomas.coranic@tuke.sk (T.C.); dominik.gojdan@tuke.sk (D.G.); jan.pasko@tuke.sk (J.P.); 2Department of Mechanical Engineering, Faculty of Technology, Institute of Technology and Business in České Budějovice, Okružní 517/10, 370 01 České Budějovice, Czech Republic; majernik@mail.vstecb.cz; 3Department of Industrial Engineering and Informatics, Faculty of Manufacturing Technologies with a Seat in Prešov, Technical University of Košice, Bayerova 1, 080 01 Prešov, Slovakia; lucia.knapcikova@tuke.sk

**Keywords:** casting mold, gating system, structural adjustments, mechanical properties of cast

## Abstract

The resulting quality of castings indicates the correlation of the design of the mold inlet system and the setting of technological parameters of casting. In this study, the influence of design solutions of the inlet system in a pressure mold on the properties of Al-Si castings was analyzed by computer modelling and subsequently verified experimentally. In the process of computer simulation, the design solutions of the inlet system, the mode of filling the mold depending on the formation of the casting and the homogeneity of the casting represented by the formation of shrinkages were assessed. In the experimental part, homogeneity was monitored by X-ray analysis by evaluating the integrity of the casting and the presence of pores. Mechanical properties such as permanent deformation and surface hardness of castings were determined experimentally, depending on the height of the inlet notch. The height of the inlet notch has been shown to be a key factor, significantly influencing the properties of the die-cast parts and influencing the speed and filling mode of the mold cavity. At the same time, a significant correlation between porosity and mechanical properties of castings is demonstrated. With the increasing share of porosity, the values of permanent deformation of castings increased. It is shown that the surface hardness of castings does not depend on the integrity of the castings but on the degree of subcooling of the melt in contact with the mold and the formation of a fine-grained structure in the peripheral zones of the casting.

## 1. Introduction

Die casting is a casting technology where molten metal is fed out of a mold loading cavity under high pressure and at high speed to a shaping cavity of a permanent mold. There it solidifies and the final cast is thus produced. The rate of a plunger acting upon the melt ranges within units of meters per second. By means of its action, the melt is fed out of the loading chamber through the gating system into the mold cavity. The transition passage between the gating system and the mold cavity is represented by a gate [1]. In the ingate, the flowing speed of a melt increases and reaches tenths of meters per second. The high flowing speed of the melt causes a rather short period of the mold cavity loading, which equals units and tenths of milliseconds. The method of the mold cavity loading allows the production of thin-walled casts in the correct shape and with high dimensional accuracy and exact copying of the surface relief of the mold cavity [2]. The final result in designing and structuring the gating systems for the casts produced by die casting technology is a cast showing adequate mechanical and qualitative properties. The gating system must assure rapid and continual loading of the shaping cavity of the mold. The correct structure of the gating systems can shorten the casting cycle duration, reduce the rejection rate, and positively influence the cast macrostructure that directly affects mechanical properties. Porosity and homogeneity correlate with strength characteristics. Final homogeneity is influenced mostly by the structure of the gate. Inside the gate, modulation of the melting occurs along with increase of flowing speed of the melt by which the shaping cavity of the mold is loaded. The mode of the mold cavity loading and the flowing speed in the gate determines the character of the final properties of the cast [3]. This article addresses the design of parts of the inlet system and the impact of modifications of individual factors on the mechanical properties of castings. The introduction presents the methodology for the design and calculation of inlet systems for castings under pressure. Using the NovaFlow&Solid program, simulations are performed to verify a suitable design solution of the inlet system to cast the electric motor flange and its shaping. The determination of the optimal geometry of the inlet notch is performed based on the assessment of the influence of the height, inlet notch on the selected mechanical properties and the porosity. At the same time, this article describes the mutual correlation of mechanical properties and porosity. It is shown that with the increasing proportion of porosity, the values of the monitored mechanical properties decrease.

## 2. Design Methodology of Gating System of Die Casting Mold for Die Casting Metals

When designing the inlet system, it is necessary to consider the technology as interconnected components of one complex. Therefore, one must focus primarily on the flow analysis of the melt in the gate. It is important to select the most suitable position for placement of the entrance and the venting system. Subsequently, it is possible to proceed to the solution and calculation of the maximum time of mold cavity filling, analysis of flowing speed of the melt in the gate, determination of the flow volume, determination of the gate dimensions, calculation of the cross section of the gating channel and determination of its shape [4,5].

### 2.1. Analysis of Flowing of the Melt in Ingate and Selection of the Most Suitable Position for Placement of the Ingate and of the Venting System

The ideal shape of the cast allows flowing of the melt inside the mold cavity along the distinct and direct paths. However, such an ideal shape can only be rarely designed, especially in gating channels and ingate. When creating, both the technical and foundry perspectives must be taken into consideration. A designer is consequently forced to search for an adequate compromise between the required and the ideal shape and suggestions and thus find a better way for the molten metal to flow. All well-known alloys utilized in the foundry industry tend to shrink during solidification and cooling. Unless this property is considered when the mold is designed, the final die-cast parts shall be defective due to shrinking occurring in the course of solidification. The defects shall have cavities in the die-cast part volume (higher porosity) and sinks of diverse sizes [6]. As the shaping mold cavity lacks risers, the high-pressure die casting represents an exception among foundry technologies. Shrinking is eliminated by resistance pressure. Therefore, the gating system must be designed so that the molten metal can transfer the force as long as possible and with minimal losses. The designer must consider pressure gradient and processes occurring inside the mold cavity, from the gate up to vents. It is convenient and applicable when the gating system is designed with the ingate placed in the dividing plane of the mold opposite the venting system. A suitable solution is to identify the pouring gate and vents so that the molten metal flowing inside the shaping mold cavity goes along the shortest trajectories. If possible, in designing the gating channels, the case of two different jets of injected metal encountering in front of the gate should be avoided. This situation is undesirable and cannot be eliminated at all times. In the case of such a situation, the ingate should be placed from inside of the die-cast part. The drawback of the structure of the central gating system is the fact that multiple cavities are not allowed and the extremely long form of the gating system causes a decrease of the speed of the molten metal flow before its entrance into the mold cavity [7,8].

### 2.2. Calculation of Maximal Time Mold Cavity Filling

The die-cast part should be designed to assure sufficient area for placement of the ingate and the venting channels [9]. The width of the ingate shall be achieved when the ingate area is divided by its height. The ingate area depends on selecting the period of the mold cavity loading and the melt flowing speed in the ingate. The mold cavity loading period is determined based on the following: (1) Thinnest walls of the die-cast part, i.e., thick walls allow longer loading periods, contrary to thin walls as those tend to get solidified prematurely. For that reason, die casting of the thin-walled die-cast parts require a shorter period of mold cavity loading. At the same time, the flowing length must be taken into consideration. If the die-cast part contains thin walls with large areas or the thin walls are placed in a considerable distance from the ingate, the period of the mold cavity loading must be shorter [10]. (2) Thermal properties of alloys and materials, i.e., temperature of liquid, range of solidification and thermal conductivity of mold material. These materials influence the period of solidification. (3) Combination of a die-cast part volume and fins, i.e., thin-walled die-cast parts, die-cast parts with a long trajectory of the melt flowing through the mold cavity and die-cast parts with special requirements regarding quality need larger fins. The condition is justified by the fact that higher metal volume can preserve the required temperature for a longer time. (4) Permitted percentage ratio of metal solidification during loading, i.e., in the case of higher surface quality requirement, it is necessary to preserve the melt with lower ratio of solidification and shorter period of the mold cavity loading. 

Maximal time of mold cavity filling t [11]:$$\;t=K.\left\{\frac{T_{i}-T_{f}+S.Z}{T_{f}-T_{d}}\right\}.T\;\left(s\right)$$

K—empirically derived constant related to mold conductivity,

T—the lowest characteristic average thickness of the die-cast part wall (mm),

T_f_—liquid temperature (K),

T_i_—melt temperature in the ingate (K),

T_d_—temperature of the mold cavity surface prior to pressing (K),

S—solidification percentage at the end of loading,

Z—conversion factors of stable units connected with the range of solidification.

The K constant gains the following values:0.0312 s/mm between steal AISI P-20 (nitrated steal) and zinc alloys,0.0252 s/mm between steal AISI H-13 (alloys of steal and chromium) and AISI H-21 (alloys of steal, chromium and wolfram) and alloys of magnesium,0.0346 s/mm between steal AISI H-13 and AISI H-21 and alloys of aluminum and brass,0.0124 s/mm between alloys of wolfram and magnesium, zinc, aluminum and brass.

Table 1 presents permitted values of material solidification depending on the wall thickness.

The Z constant gains the following values:4.8 °C/% for alloys of aluminum ASTM 360, 380 a 384, all sub-eutectic alloysAlSi (Cu/Mg) containing less than 12% silicium,5.9 °C/% for alloys of aluminum ASTM 390, supra-eutectic alloys AlSi (Cu/Mg),3.7 °C/% for magnesium alloys,3.2 °C/% for zinc alloys,4.7 °C/% for brass [12].

### 2.3. Calculation of Flowing Speed of the Melt in the Ingate

The flowing speed of the molten metal in the ingate influences the mechanical properties of the die-cast part and the quality of its surface. New high-pressure die casting machines can generate a speed of up to 100 m·s^−1^, yet degradation of the mold commences at approximately 40 m·s^−1^. Thus, choosing the speed within the range from 40 up to 100 m·s^−1^ is rather impractical. Porosity caused by the bonding of gas in the die-cast part volume can be decreased without extreme increase of speed by designing the gating system and ingate so that avoidance of shocks and consequent reversible flowing and mixing of the melt is assured. Flowing of the melt through the gating system must be continuous. The reversible flowing effort can be made when the trajectory of melt flowing contains lugs, sharp direction changes or incorrectly reduced diameters [13]. Table 2 presents recommended values of the flowing speed of the melt in the ingate. 

The flowing speed of the melt in the ingate v_1_ can be determined according to the following formula: $$v_{1}=\frac{m_{c}}{p.t.d_{ch}.0.785}\left(m.s^{-1}\right)$$

*m_c_*—the weight of cast (kg),

*p*—density of alloy (kg·m^−3^),

*d_ch_*—diameter of filling chamber (m).

### 2.4. Determination of the Flow Volume

The flow serves as a heat accumulator and as a tank of low-quality oxidized metal. The flows are necessary for thin walls of the die-cast part or if the die-cast part must get solidified at a higher temperature [14]. An example is the circumfusing of the cores placed at a considerable distance from the ingate. The melt flows around the core through narrow walls from both directions, and adequate temperature must be assured to achieve a unified and firmly joined bond. Table 3 presents recommended flow volumes for conventional die casting machines depending on the lowest thickness of the wall.

### 2.5. Determination of the Ingate Dimensions

Dimensions of the ingate depend on the method of connection of the ingate to the cast. Figure 1 shows a scheme of the method of projection of the ingate connection to the cast with cylindrical area.
$$A_{I}=\frac{G}{\rho .t.v_{I}}\;\left(m^{2}\right)$$

*G*—sum of weight of die-cast part and of the weight of flows (kg).

Ingate length a:$$a=\frac{2.\rho .R.\alpha }{360}=\frac{2.\pi .R.60}{360}=\frac{\pi .R}{3}\;\left(m\right)$$

R—die-cast part radius (m).

Ingate height b:$$b=\frac{A_{I}}{a}\;\left(m\right)$$

### 2.6. Calculation of the Cross Section of the Gating Channel and Determination of Its Shape

The section of the gating channel is trapezoidal, with the slope of the walls ranging between 10°–15°. The channel height to width ratio should vary within the scale from 1:1 up to 1:3. The standard ratio to be selected is 1:2. Calculation of the channel cross section depends on the diversity of the mold. The areas of the cross sections of the channels are mainly influenced by the branching of the media [15]. When the channel is divided into a branching, its total cross section should be increased by 5–30% after each dividing in the direction from the ingate towards the tablet. The procedure in calculation begins with the design proposal of diameters of the channels in the direction from the ingate. Area of the gating channel A:$$A=n.2.A_{I}\;\left(m^{2}\right)$$

Cross section of the gating channel is show in Figure 2.

Height of the gating channel CT:$$A=\mathrm{CB}.\mathrm{CT}-{\mathrm{CT}}^{2}.\mathrm{tg}\left(90{}^{\circ}-\alpha \right)=2.{\mathrm{CT}}^{2}-{\mathrm{CT}}^{2}.\mathrm{tg}\left(90{}^{\circ}-\alpha \right)$$
$$\mathrm{CT}=\sqrt{\frac{A}{2-tg\left(90{}^{\circ}-\alpha \right)}}\;\left(m\right)$$

*α*—angle of wall inclination of the gating channel [°].

Width of the gating channel CB:CB=2.CT (m)

## 3. Materials and Methods

### 3.1. Characteristic of Monitored Parameters

From the point of view of the influence of the structure of the gating system of die casting mold upon the die-cast part quality, the most significant parameters affect the loading mode of the die casting mold cavity and type of the melt flowing through the gating channels. These can be considered as follows: Cross section of the ingate and molding of the die-cast part inside the mold cavity regarding the placement of the cores in the mold shaping cavity. Molding of the die-cast part and placement of the cores influence the qualitative properties of the die-cast part. It is necessary to ensure such placement of the cores so that the flow of the melt passing through the mold shaping cavity avoids direct hitting of the cores. Formation of contractions, cold laps or weld lines would be eliminated [16]. Both the sectional area of the channels, which is in the analytical design frequently represented by the medium width of the gating channel, and its shape influence the flowing in the gating channels and the melt flowing speed in the gating channel. Firstly, the cross section of the ingate depends on the shape of the mold cavity and on the ingate height. In the ingate the cross section becomes smaller by which the speed of the melt jet increases. Consequently, the mold shaping cavity is loaded under the melt. The ratio between the areas of the ingate and of the die-cast part is determined by the loading mode of mold shaping cavity. According to the methodology of the ingate structure described, the ingate width is constant, in our case a=60.968 mm, for the formation of the gating channel surface; the values of the gating channel height were determined as follows [17,18]:

b1 = 1.25 mm,

b2 = 1.03 mm,

b3 = 0.92 mm,

b4 = 0.82 mm,

b5 = 0.75 mm.

Lower and upper values of the ingate height were determined by means of simulation program NovaFlow&Solid through the module NovaShot, which determined maximal and minimal gating channel height for particular die-cast parts. Other values were determined to provide diverse loading modes of the mold shaping cavity [19,20].

### 3.2. Structural Design of the Gating System

In solving the design of the gating systems, we used drawing documentation of the cast of the electric motor flange. On the basis of this drawing documentation, the model shown in Figure 3 was generated in the CAD program Pro/Engineer-Creo Parametric 2.0 3D.

By utilizing the respective modules in the program, the cast volume was determined, and based on the density of alloy which shall be used for casting, the cast weight was calculated. According to the weight and volume characteristics presented in Table 4, the calculation of the dimensional factors of the mold gating system was realized [21].

### 3.3. Structural Design of the Gating Channels and the Ingate

Before we designed the gating system, it was necessary to consider the limits, ensuring the type of the die casting machine, from the mold size to the shaping liners [22]. Dimensions of the gating system are presented in Table 5. The shape of the channels, placement of the shaping cavities and the cast molding methods in the mold are shown by the 3D model of the gating system designed using the numerical solution in Figure 4.

### 3.4. Cast Forming

The method of the cast forming in the mold represents a significant aspect of the design part of the gating system structure. For the respective gating systems, two variants of forming were proposed. Figure 5 shows the cast in the case where the melt jet is directed through the ingate so that it avoids hitting the cores, determining the structural holes of the model. The variant shown in Figure 6 represents the case in which the melt jet is directed right towards the core. The formation of the melt jet passing through the mold shaping cavity allows the elimination of defects occurring due to incorrect forming during the designing phase [23].

## 4. Results

### 4.1. Simulation Test of the Cast Forming

During simulation, the development of the melt jet formation was observed in the mold shaping cavity. In the case of the variant with the melt jet directed off the cores, the jet remained compact and followed its direction as spinning of the melt and consequent closure of gases and air in the melt volume was avoided [24]. The shape and formation of the melt jet are shown in Figure 7. The hitting of the melt jet against the core resulted in splitting the melt jet into two independent flows, as shown in Figure 8.

In gradual loading of the mold shaping cavity, the two partially solidified melt jets are joined behind the core, leading to imperfect joining in the cast volume, the occurrence of shrinks and the formation of cold laps and weld lines. Due to the facts mentioned above, directing the melt jet without hitting the cores appears more advantageous. Figure 9 shows the distribution of defects occurring in the cast body in the case where the melt jet hit the core [3,25].

### 4.2. Simulation Test of Suitability of the Selected Structural Design

The 3D model of the gating system, which was subjected to simulation tests, focused on verification of suitability of the selected structural design. The simulations were performed with the NovaFlow&Solid program. In the simulation, the observed determining parameter of the design suitability was the ratio of shrinks in the cast body and the gating channels after solidification. Figure 10 shows a simulation of the gating system and distribution of defects after completion of the casting cycle [26].

The simulation and visual representation proved that the numerical design proposal of the gating system was not suitable. The shrinks in the cast body represented 5.4% of the total design. The simulation analysis detected that the formation of the shrinks begins at a time of t = 0.042 s. At this time, the mold shaping cavity is 100% loaded, and the influence of the resistance pressure, which eliminates the formation of shrinks, should be visible. However, such situations, as well as the resistance phase, are absent. To clarify the state, it was necessary to focus on the narrowest spot of the gating system represented by the ingate. Examination of the temperature development proved that right at the time of t = 0.042 s, the melt temperature in the ingate drops below the value of melting (Figure 11), the ingate solidifies and prevents resistance pressure from influencing the mold shaping cavity. One of the causes of the state is the structure of the channels. There exists a presumption that the main and the side gating channels dispose of considerably large areas in the cross section. The melt loses pressure and speed, resulting in premature cooling and thus solidifying the ingate. Resistance pressure loses its effect and can be observed as shrinks due to material shrinkage when transiting from the liquid to the solid state [27].

### 4.3. Adjustment of Structural Design

#### 4.3.1. Adjustment of Structural Design of the Gating Channels

Diminishing the areas of sections of the gating channels with the preservation of technological parameters shall increase the speed of melt flow and mold shaping cavity loading within a shorter period. With the parameters mentioned above, the premature solidification of the ingate shall be avoided. The shape of the gating system, as well as the method of cast molding, remained alike. The most suitable dimensions of the gating channels are presented in Table 6.

The adjustment of gating channels resulted in diminishing the area of the main channels by 38.68% and of the side channels by 34.33%. The weight of the overall gating system was decreased by 10.58%.

#### 4.3.2. Simulation Test of the Adjusted Structural Design

Simulation tests of the adjusted structural design proved the prediction that diminishing the section of the gating channels results in faster loading of the mold shaping cavity. The possibility of premature solidification of the melt in the ingate is eliminated. Figure 12 shows simulation of the gating system after completion of the casting cycle. The visual illustration does not show any visible defects regarding the compactness of the cast [28,29].

### 4.4. Analysis of Mechanical Properties

In designing and in the internal operation of the product, it is necessary to know the material properties that do not refer only to the chemical composition of material but also to its inner structure, mechanical properties and mutual relationships among the properties. The acquired knowledge allows assessing behaviour of the individual materials in operation when being stressed and consequently to specify data and documents for dimensioning. In the case examined in the paper, the pressure tests were performed after assessment of selected specific and mechanical properties of die-cast part was performed. The assessed properties included the value of permanent deformation “s” and the surface hardness according to Brinell “HB” of the die-cast part.

#### 4.4.1. Chemical Composition of Experimental Meltage

Experimental samples (flange of the alternator) have been cast using aluminium alloy EN AC 47100, whose chemical composition is given in Table 7 according to the EN 1706 standard.

#### 4.4.2. Analysis of Permanent Deformation

Tests of permanent deformation were performed with testing machine TIRAtest 28,200 (TIRA GmbH, Shalkau, Shalkau, Germany). Deformation was examined in the case of a particular critical place of the die-cast part (Figure 3). According to the assessment of the theoretical basis of the hydrodynamics of the liquid melt flow when bypassing the cores, the place was considered to be a critical one. The measurement was performed in compliance with the GME 06 007 and GME 60 156 standards. The GME 06 007 standard describes the experimental procedure during test performance and the GME 60 156 standard describes the method of assessment of the test of permanent deformation. According to the prescribed values related to the standard, the permanent deformation reaches the value of 0.150 mm. Table 8 presents the importance of permanent deformation concerning the change of the ingate height. Figure 13 shows the relationship among average values of permanent deformation and modification of the ingate height.

#### 4.4.3. Analysis of Hardness

According to Brinell “HB“, values of hardness were measured in the case of five places of the selected assessed samples with the change of the ingate height. The results are shown in Table 9 and graphical development can be seen in Figure 14.

#### 4.4.4. Analysis of Internal Homogeneity

The places for evaluating the homogeneity of the castings were chosen concerning the possibility of assessing the mutual correlation of the homogeneity of the castings and the permanent deformation. X-ray analysis was performed using a VX1000D X-ray machine (North Star Imaging, Marlborough, Marlborough, MA, USA). Figure 15 presents an X-ray image of the critical point of the casting on sample 4.C, which belongs to the set of castings showing the lowest values of permanent deformation. No internal defects are apparent in Figure 15. Figure 16 and Figure 17 present X-rays of samples belonging to a set of samples showing the highest permanent deformation values. Figure 16 and Figure 17 clearly show brighter spots, representing internal inhomogeneity.

#### 4.4.5. Analysis of Microstructure

The fine-grained structure of the castings in the area of contact of the melt with the face of the mold (Figure 18) is observable in all samples analyzed. The transition of the fine crystals into a coarser structure is smooth, with the thickness of the fine-grained layer being in the order of a few μm to 1 mm. The formation of a fine-grained structure can be explained by the high degree of subcooling of the melt in contact with the relatively cold face of the mold. The degree of subcooling depends mainly on the melt temperature, the mold temperature, the method of molding the casting and the mode of filling the mold cavity.

## 5. Discussion

The quality of a die-cast part is a property influenced by several factors, including selecting the casting machine through the setting of technological parameters and suitable structure of the gating system up to its parts. The gating system must assure fast and continuous mold cavity loading. This is determined by good mechanical properties and inner soundness of the die-cast part. The correct mold cavity filling was performed by simulation software of NovaFlow&Solid (Version 6.5) in the analytical design of the gating system after completion of structural modification of channel areas and die-cast part forming. The assessed parameter was the percentage proportion of contractions in the die-cast part volume in simulation tests. During the preparation of the submitted paper, the assessment of die-cast part quality characterized by mechanical properties was performed through a selection of representative parameters such as die-cast part hardness “HB” and permanent deformation “s” of critical place of the die-cast part with the change in the ingate height. RTG analysis and permanent deformation were performed on the basis of the anticipated mutual correlation of these parameters.

### 5.1. Assessment of Simulation Tests

The influence of the cast forming in the mold cavity was examined through simulation directing the melt jet out of the ingate directly to the core (Figure 6) and directing the melt jet off the cores (Figure 5). In an analysis of the melt jet direction, the cast was connected to the adjusted gating system, which showed a zero percentage ratio of shrinks when the melt jet did not hit the cores directly. In the case of the melt hitting the core, the defects shown in Figure 9 occurred behind the core and the ratio of contractions reached the value of 2.8%. Simulation tests verified the structural design proposal of the gating system. The ratio of contractions in the cast body reached the value of 5.4%. The localization of defects is shown in Figure 10. As demonstrated and consequently proved by the simulation, the situation was caused by a large area of the channels. The loss of speed and pressure could be observed in the channels, resulting in slow mold cavity loading; the state led to premature solidification of the ingate and inhibition of resistance pressure influence. By reducing the dimensions of channels, the desired mold cavity loading was achieved without showing any percentage ratio of shrinks. At the same time, the overall time of the mold cavity loading during the period of complete cast solidification decreased from the value of 21.567 s to the value of 20.877 by means of which higher work efficiency can be achieved, which shall be reflected in a higher number of casting cycles per unit of time.

### 5.2. Assessment of Permanent Deformation

The lowest values of permanent deformation were detected with the samples produced from the die-cast parts cast with the ingate height of 0.82 mm. Based on observation, it can be assumed that the values of permanent deformation depend on the ingate height due to modulation of jet and speed of the melt flowing through the ingate, which consequently determined the mold cavity loading mode. The conditions for the aforementioned parameters are that in the case where the ingate height is 0.82 mm, the melt jet acquires the speed which forms the mold cavity loading mode in a combination of turbulent and disperse flowing.

### 5.3. Assessment of Hardness

Measurement of the selected die-cast parts was performed at five places. The substantial difference between measured hardness values dependent on the change of the ingate height within the frame of the performed analysis has not been proved. Acquired values have also confirmed the fact. The presented facts demonstrate precondition, i.e., die casting is characterized by a high level of the melt subcooling in contact with the relatively cool wall of a mold, which results in the formation of a fine-grained structure on the peripheral walls of the casting. Determining the factors of HB hardness values in die casting is the generation of primarily the formed crust with a fine-grained structure, as documented in Figure 18.

## 6. Conclusions

The presented paper analyzed the problem of the gating system in the case of a particular die-cast part. The influence of the individual parameters upon the melt flowing using the selected structural design impacting the die-cast part quality was examined. The specified mechanical properties and homogeneity were examined. Permanent deformation “s” and surface hardness “HB” were selected out of the mechanical properties. The die-cast part homogeneity was assessed during the designing stage based on the percentage proportion of the contractions and during the experimental phase according to RTG analysis of the selected die-cast part groups. The achieved results proved that the ingate height represents one of the basic structural factors influencing qualitative properties of the die-cast parts and determining the speed and the mode of the mold shaping cavity loading. In producing test samples from alloys of identical composition, the structural parameters are influenced by the rate of the melt solidification and by resistance pressure. As the examination was focused on the influence of the structural design of the gating system upon the quality of die-cast parts, all of the technological parameters, including the melt temperature, mold temperature, press ram speed in the loading chamber and resistance pressure, were maintained at a constant level and continuously monitored. We preconditioned practically identical structures in the production of test samples in the case of each working cycle. This was also confirmed by the absence of a distinct difference in hardness of the individual samples. The analyses of dimensions, structure and location of inner defects (RTG analysis) allowed determining the correlation of mechanical characteristics and these parameters. Firstly, it was proved that the value of permanent deformation “s” corresponds with the size of the cavities reducing the cross section of the die-cast part. Apart from their size, the location of the cavities in the die-cast part volume is also significant. It can be stated that evenly distributed cavities of smaller sizes reduced mechanical properties in a smaller proportion contrary to the cavities with relatively large sizes or clusters of cavities in case of maintenance of the identical volume proportion of cavities in the die-cast part volume. The conditions of the mold loading on cavity volume were distributed in the sample. It can be assumed that following the setting of technological parameters of die casting and selecting suitable alternatives of the gating system design are supported by simulation methods during the designing stage. By additional correction during production, this phase leads to producing high-quality die-cast parts meeting operation requirements related to their function.

## Figures and Tables

**Figure 1 materials-14-03755-f001:**
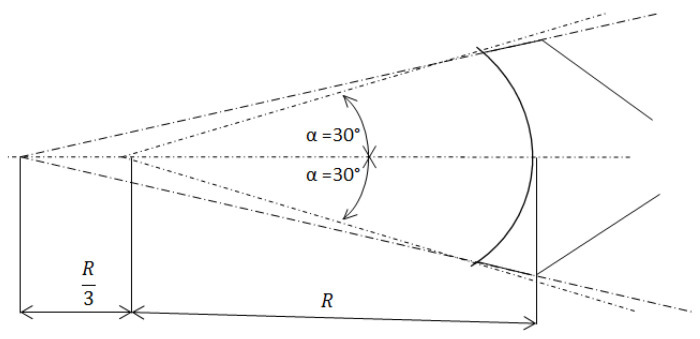
Scheme of attachment of the ingate to the cylindrical area ingate area A_I_.

**Figure 2 materials-14-03755-f002:**
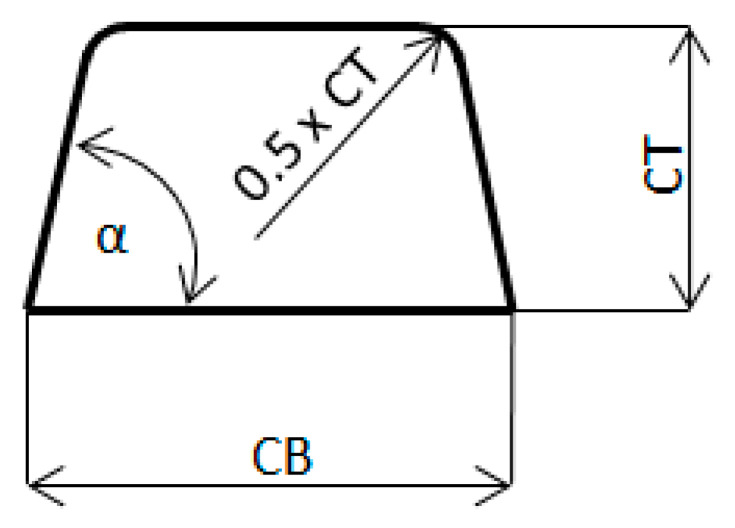
Cross section of the gating channel.

**Figure 3 materials-14-03755-f003:**
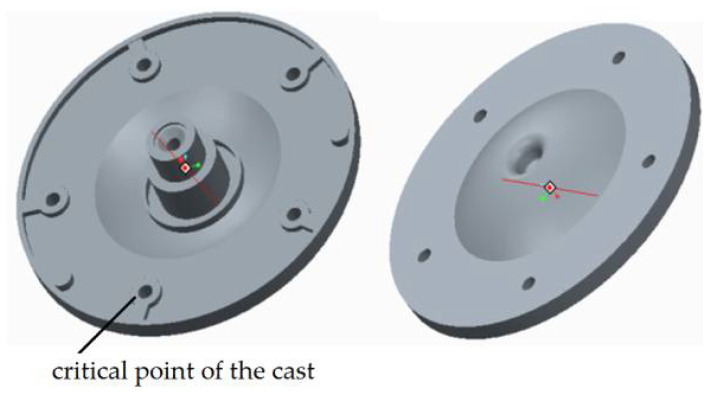
3D model of electric motor flange.

**Figure 4 materials-14-03755-f004:**
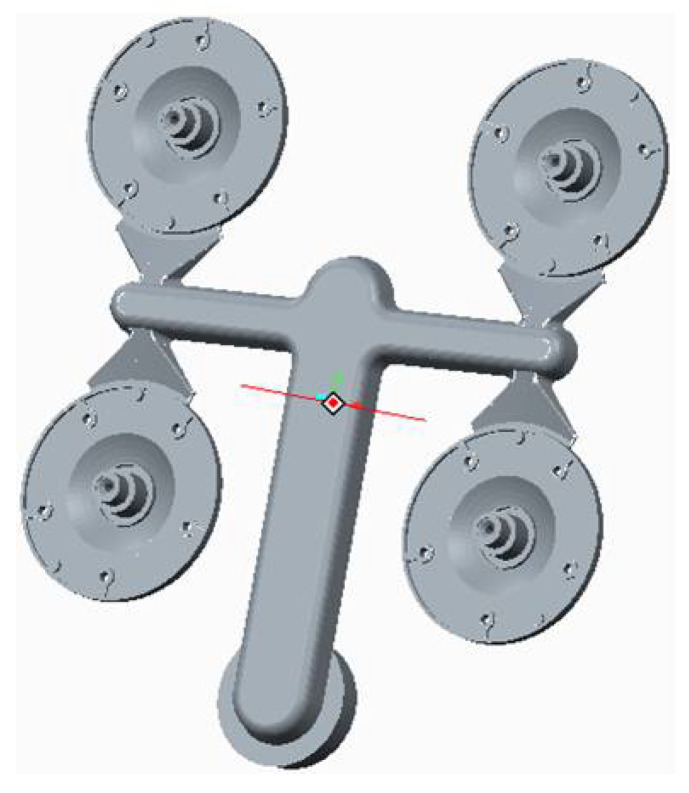
3D model of designed gating system.

**Figure 5 materials-14-03755-f005:**
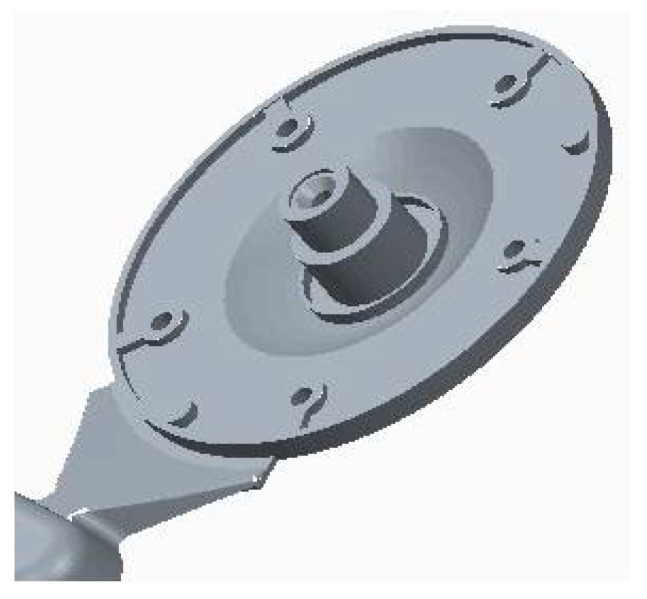
The melt jet is outside the core.

**Figure 6 materials-14-03755-f006:**
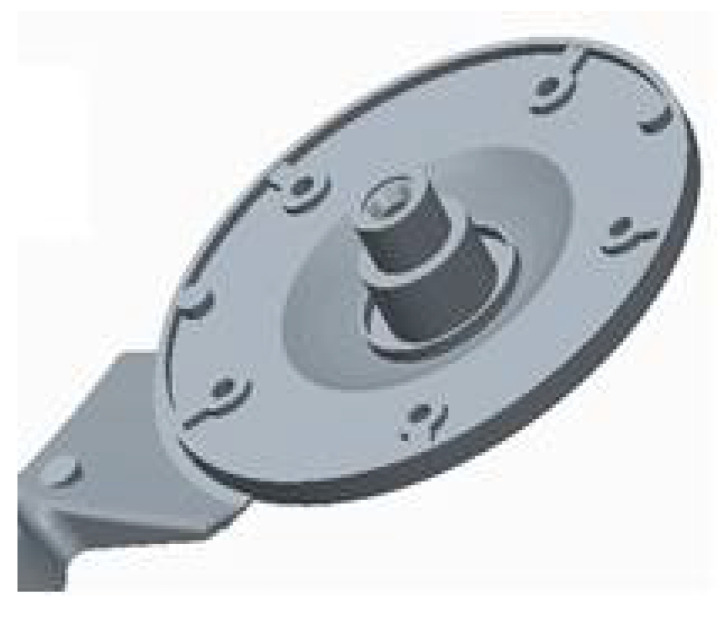
The melt jet is directed right towards the core.

**Figure 7 materials-14-03755-f007:**
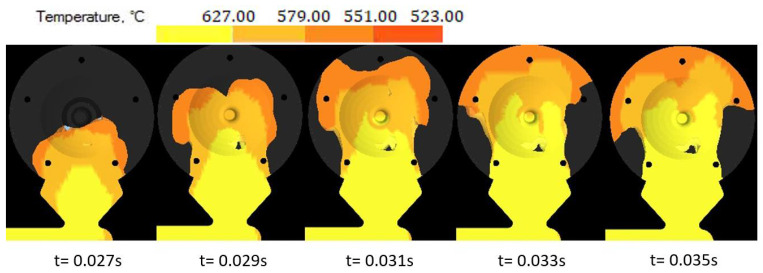
Formation of the melt jet outside the cores.

**Figure 8 materials-14-03755-f008:**
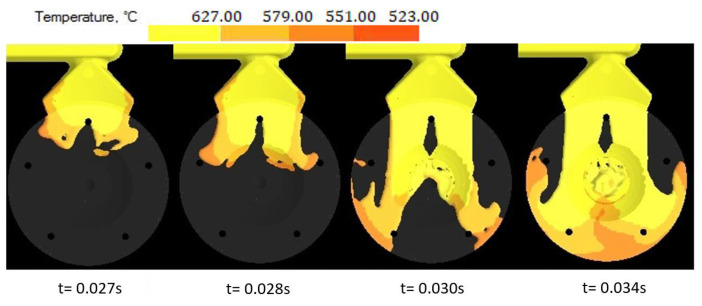
Formation of the melt jet when hitting against the core.

**Figure 9 materials-14-03755-f009:**
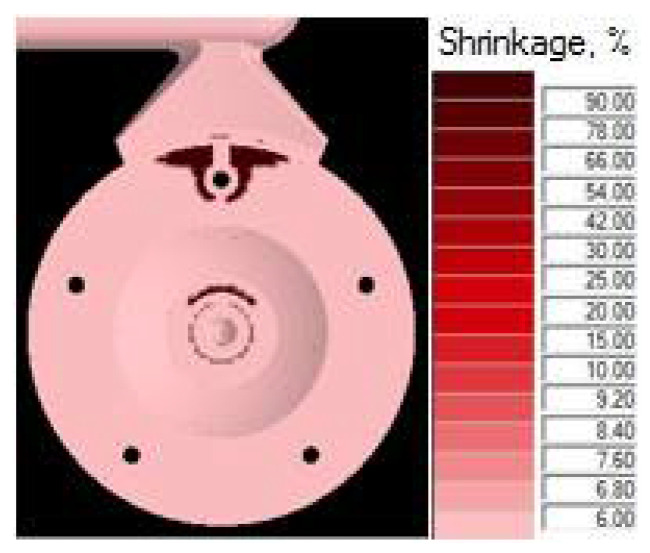
Distribution of defects for incorrect of cast forming—contractions 2.8%.

**Figure 10 materials-14-03755-f010:**
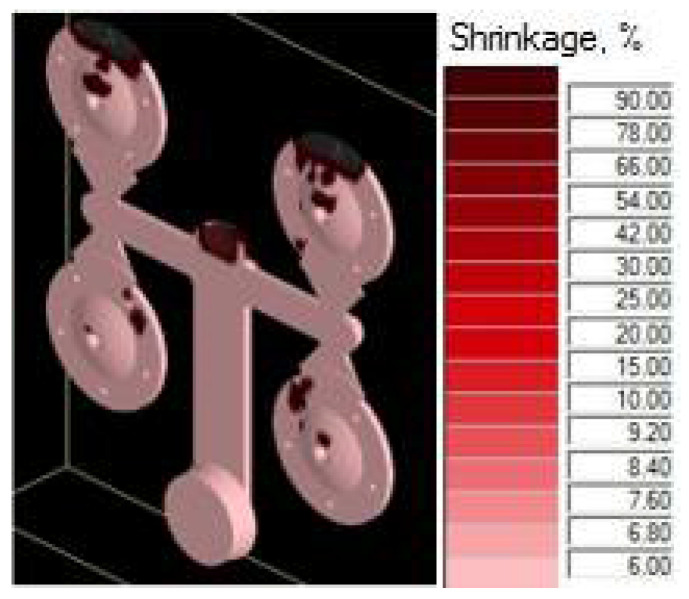
Distribution of defects in gating system—contractions 5.4%.

**Figure 11 materials-14-03755-f011:**
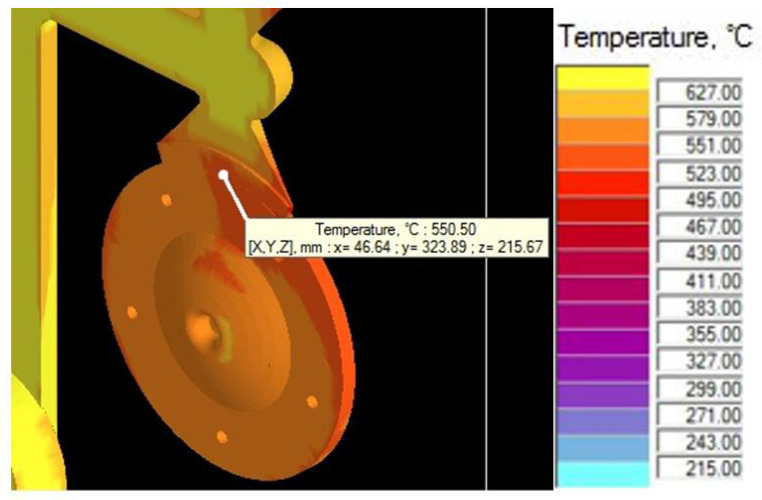
Temperature in the ingate.

**Figure 12 materials-14-03755-f012:**
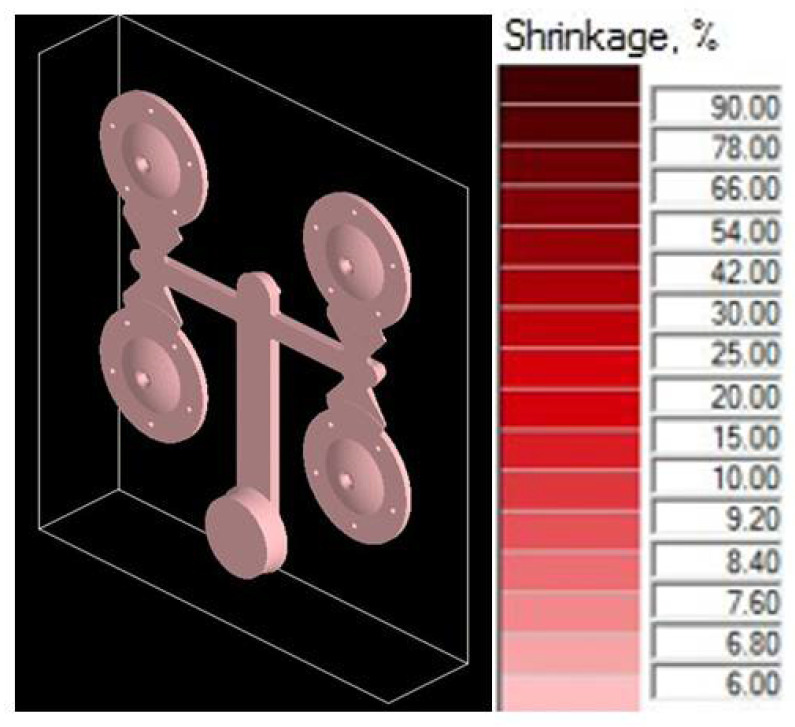
Distribution of defects in adjusted gating system—contractions 0%.

**Figure 13 materials-14-03755-f013:**
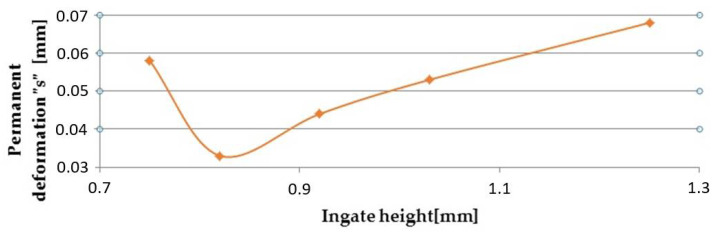
Relationship between permanent deformation and change of the ingate height.

**Figure 14 materials-14-03755-f014:**
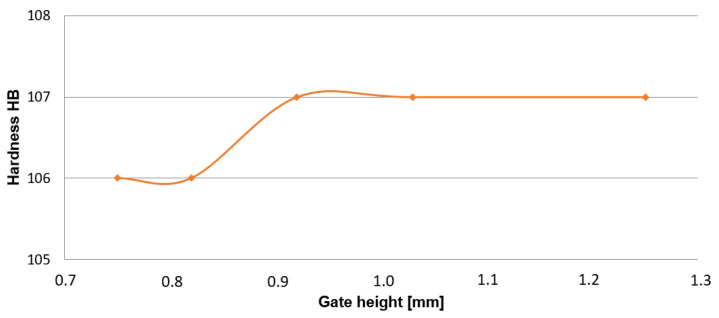
Relationship between the hardness of the diecast part HB and change in the ingate height.

**Figure 15 materials-14-03755-f015:**
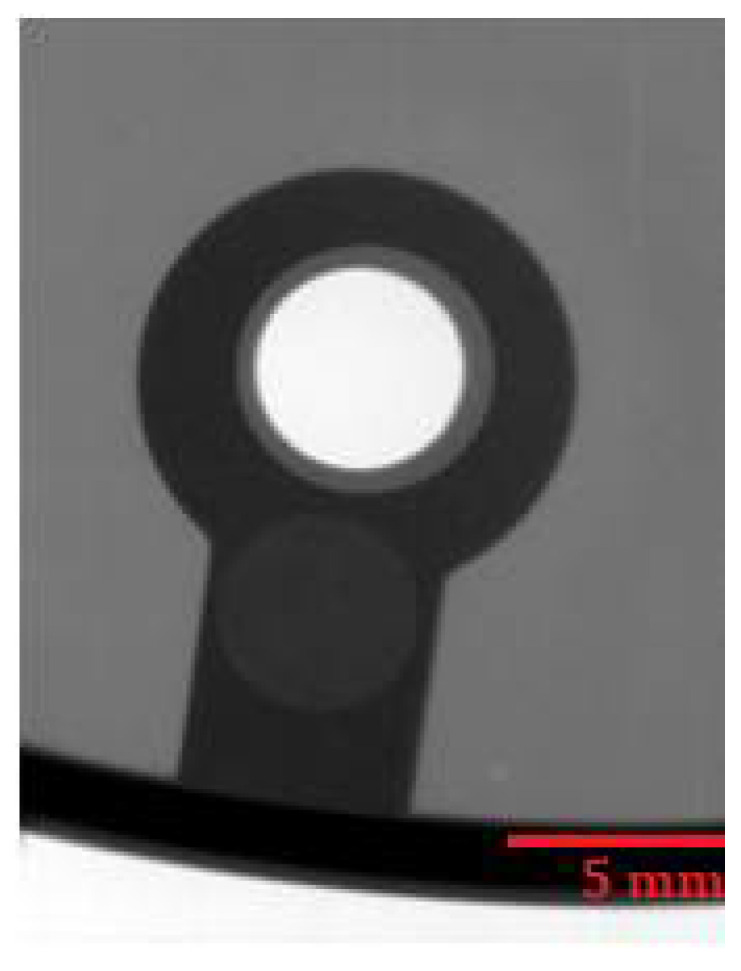
RTG—Sample 4.C.

**Figure 16 materials-14-03755-f016:**
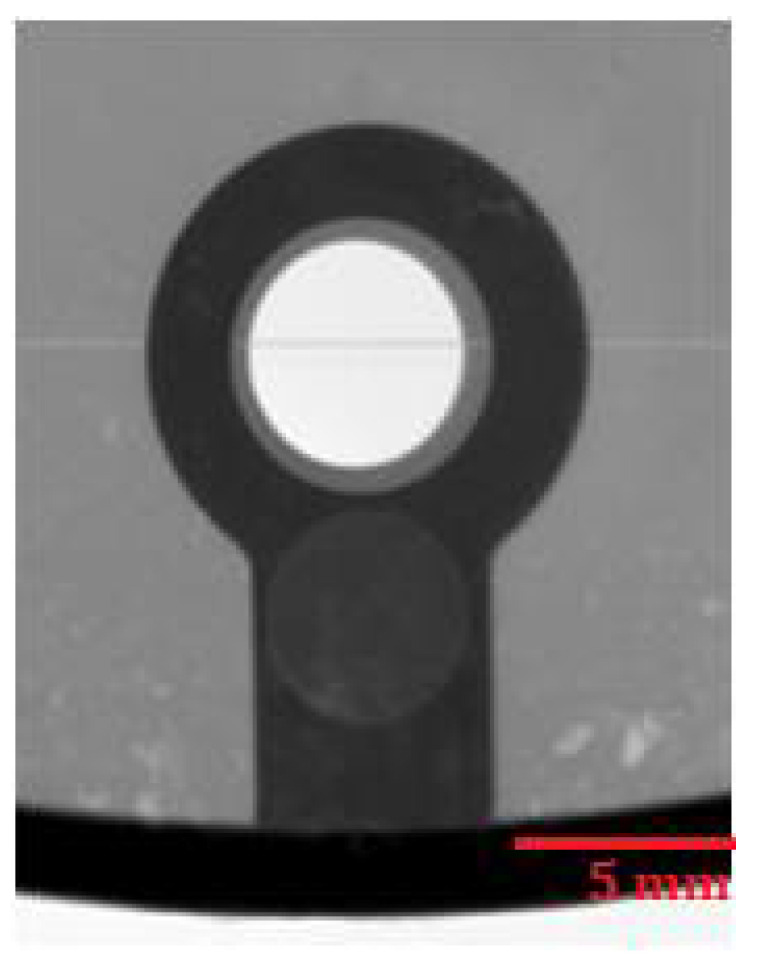
RTG—Sample 1.B.

**Figure 17 materials-14-03755-f017:**
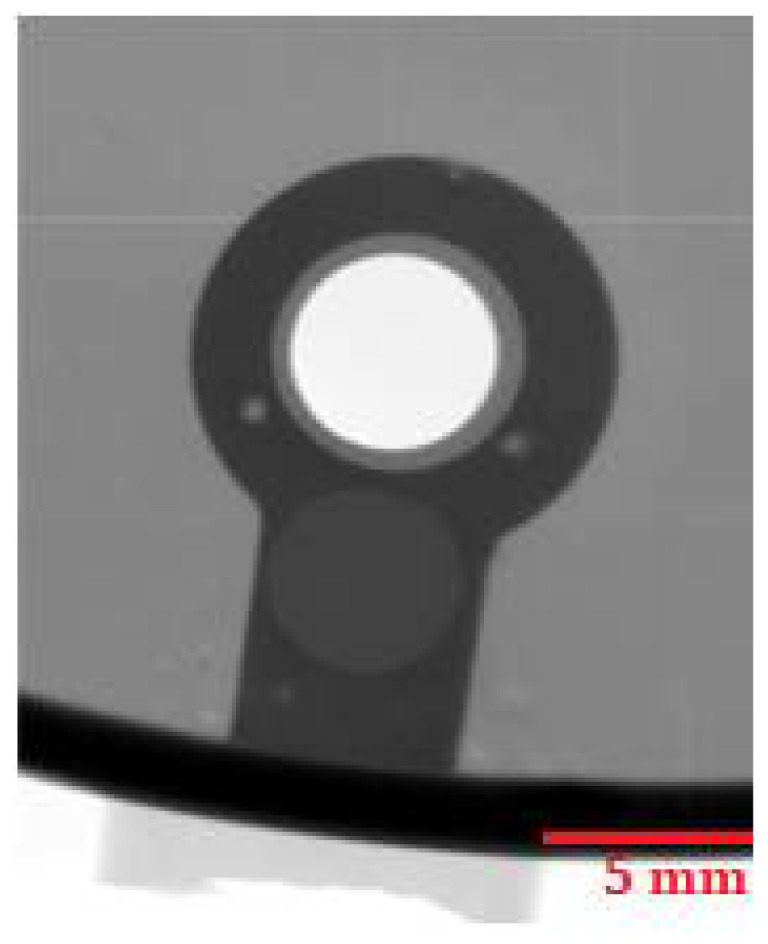
RTG—Sample 2.C.

**Figure 18 materials-14-03755-f018:**
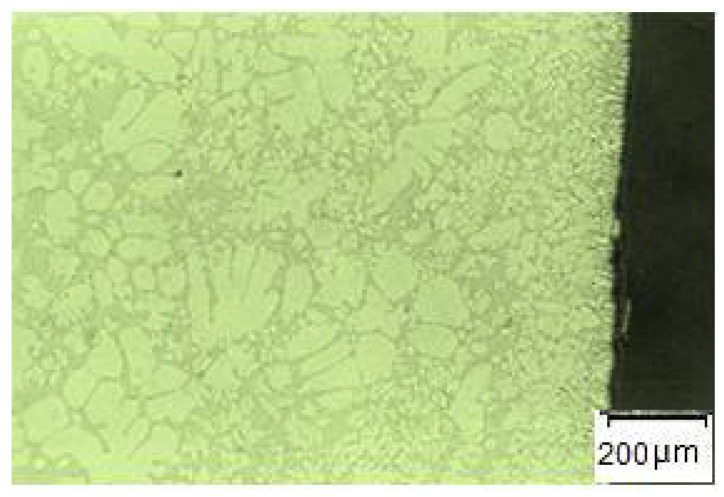
Microstructure of the edge part of the casting.

**Table 1 materials-14-03755-t001:** Permitted values of material solidification depending on the wall thickness.

Wall Thickness [mm]	Permitted Values of Material *S* (%)		
	Aluminium	Magnesium	Zinc
<0.8	5	10	5–15
0.8–1.25	5–25	5–15	10–20
1.25–2	15–35	10–25	15–30
3–2	20–50	20–35	20–35

**Table 2 materials-14-03755-t002:** Recommended values of the flowing speed of the melt in the ingate.

Type of Alloy	Recommended Values of the Flowing Speed of the Melt in the Ingate (m·s^−1^)	
	Standard Casting	Vacuum Casting
Aluminium	20–60	15–30
Zinc	30–50	
Magnesium	40–60	
Copper	20–50	

**Table 3 materials-14-03755-t003:** Recommended volumes of flow.

Characteristic Wall Thickness of a Gate Segment (mm)	Flow Volume, Percentage Ratio Out of Segment Volume	
	For High Surface Quality	For Lower Surface Quality
0.90	150%	75%
1.30	100%	50%
1.80	50%	25%
2.50	25%	25%
3.20	-	-

**Table 4 materials-14-03755-t004:** Weight and volume characteristic of die-cast part.

Quantity	Value
Alloy	EN AC 47100—AlSi12Cu(Fe)
Die-cast part volume	51,697.9 × 10^−9^ m^3^
Alloy density	2650 kg·m^−3^
Die-cast part weight	0.136 kg

**Table 5 materials-14-03755-t005:** Dimension of the gating system.

The Main Gating Channel		The Side Gating Channel	
Parameter	Value	Parameter	Value
Area/A	811.36 × 10^−6^ m^2^	Area/A	368.8 × 10^−6^ m^2^
Height/CT	14.59 × 10^−3^ m	Height/CT	14.59 × 10^−3^ m
Weight/CB	51.70 × 10^−3^ m	Weight/CB	29.18 × 10^−3^ m
Length/L	264 × 10^−3^ m	Length/L	280 × 10^−3^ m
Ingate			
The area/A_I_		75.97 × 10^−6^ m^2^	
Length/a		60.968 × 10^−3^ m	
Height/b		1.25 × 10^−3^ m	

**Table 6 materials-14-03755-t006:** Adjusted dimensions of gating channels.

The Main Gating Channel		The Side Gating Channels	
Parameter	Value	Parameter	Value
Area/A	497.5 × 10^−6^ m^2^	Area/A	242.19 × 10^−6^ m^2^
Height/CT	14.59 × 10^−3^ m	Height/CT	14.59 × 10^−3^ m
Weight/CB	38.0 × 10^−3^ m	Weight/CB	20.5 × 10^−3^ m
Length/L	264 × 10^−3^ m	Length/L	280 × 10^−3^ m

**Table 7 materials-14-03755-t007:** Chemical composition of experimental meltage of used alloy % content of elements.

Chemical Composition of the Experimental Cast of the Applied Alloy % of Elements Content											
Al	Si	Fe	Cu	Mn	Mg	Cr	Ni	Zn	Pb	Sn	Ti
85.50	11.82	0.76	1.03	0.27	0.18	0.03	0.12	0.30	0.03	0.03	0.03

**Table 8 materials-14-03755-t008:** Values of permanent deformation “s” in relation to change of the ingate height.

Sample No.	Gate Height (mm)	Gate Weight (mm)		Permanent Deformation “s” (mm)
				Arithmetic Average
1.A	1.25	60.968	0.064	0.066
1.B			0.068	
1.C			0.065	
2.A	1.03		0.048	0.053
2.B			0.057	
2.C			0.055	
3.A	0.92		0.042	0.044
3.B			0.044	
3.C			0.047	
4.A	0.82		0.035	0.033
4.B			0.033	
4.C			0.032	
5.A	0.75		0.057	0.058
5.B			0.054	
5.C			0.062	

**Table 9 materials-14-03755-t009:** Values of the surface hardness of the die-cast part “HB” in relation to change of the ingate height.

Sample No.	Gate Height (mm)	Gate Weight (mm)	Measurement					Arithmetic Average
			No. 1	No. 2	No. 3	No. 4	No. 5	
1	1.25	60.968	108 HB	107 HB	108 HB	107 HB	107 HB	107 HB
2	1.03		109 HB	107 HB	106 HB	107 HB	106 HB	107 HB
3	0.92		106 HB	108 HB	107 HB	108 HB	107 HB	107 HB
4	0.82		106 HB	104 HB	107 HB	107 HB	107 HB	106 HB
5	0.75		105 HB	107 HB	105 HB	106 HB	107 HB	106 HB

## Data Availability

Not applicable.

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
