# Peer review of "Influence of Gating System Parameters of Die-Cast Molds on Properties of Al-Si Castings"

_materials, 2021, doi:10.3390/ma14133755_

Round 1

Reviewer 1 Report

The authors discussed the detailed processes for gating system design for Al-Si alloy high pressure die casting, which is important for current industrial applications. This paper may gain strong interest from the readers; however, much improvement is needed to meet the quality of Materials. The details are listed below:

1) Since this is a manuscript rather than a powerpoint talk, item listing shall be converted into logical paragraphs. Line 46-55 on page 2 shall be rewritten properly. Please change other parts similarly.

2) For equations, it shall be led by a proper sentence, followed by mathematical equations, and notation explanation. Symbols cannot be simply listed (Line 109-125, page 2), and it shall be started with where. For example, where K is empirically derived constant related to mould conductivity.

3) Several typos shall be corrected, for sample vI on line 159 (page 4).

4) Tables shall be adjusted properly (Table 7), and one table can only have a heading. Change “the rest” to “Al Bal.”, and remove “according to EN 1706”,…

Author Response

The answers to the reviewer's comments are in the attached document. 

Reviewer 2 Report

In injection molding, the main parameters are: dimensional accuracy, surface roughness, strength and tightness. Features of injection molding:

  1. The high kinetic energy of the moving melt and the pressure in the pressing make it possible to obtain castings with low roughness (this point is not reflected in the manuscript).
  2. Gas-tight shape and lubricant promote the formation of gas-air porosity in castings. This determines the mechanical properties and tightness of the casting (the manuscript does not include this effect of the lubricant).
  3. The high intensity of thermal interaction between the melt and the mold leads to the formation of a fine-grained structure in the surface layers of the casting. This increases the strength of the casting and the productivity of the process. (In the manuscript, instead of general phrases about the structure of surface layers, it is necessary to indicate specific values, namely the minimum and maximum thickness of a layer with fine grain, depending on the mode of filling the form with photographs of these layers. In addition, the performance of the process should be given).
  4. Until the moment of solidification of the feeder, the reduction of shrinkage porosity is effectively influenced by the action of the press piston on the melt (the manuscript does not contain detailed data on the period of effective action of the pre-pressing on the melt).

The article is aimed at reducing the gas-air and shrinkage porosity in the aluminum casting of the electric motor housing by selecting the optimal mode for filling the mold. At the same time, it is not clear how relevant this aspect is for this casting. Is the casting subject to heat treatment? Does it buckle and leak after heat treatment?

Instead of a detailed computer analysis of the effective period of pre-pressing at different parameters of the feeder for the supply option (Fig. 5), the authors consider a deliberately ineffective option with filling the mold through the rod (Fig. 6).

The manuscript needs to be reworked from the standpoint of reducing the gas-air and shrinkage porosity of the body aluminum part by selecting an effective mode for filling the mold by injection molding.

When revising the article, you need:

  1. In the introduction, indicate the relevance and purpose of the article.
  2. Screenshots of computer simulation with scales.
  3. Describe in detail the method for determining residual deformations, stresses and indicate the points of hardness measurement.
  4. On the graph of the change in hardness, change the scale of the scale so that the difference (slope of the line) is clearly visible.
  5. On the photo of microstructures and castings, indicate the scale.
  6. In the annotations and conclusions on the work, provide quantitative indicators of technology improvement.

Author Response

(The authors gave the same response as above.)

Round 2

Reviewer 1 Report

I am fine with current manuscript although English writing can still be improved.

Reviewer 2 Report

In Figure 18, you must specify the scale.